# Differential Sandwich-Type Results for Symmetric Functions Connected with a *Q*-Analog Integral Operator

**Sheza M. El-Deeb** [1,*,†,‡] [ID] **, Teodor Bulboacă** [2] [ID] *and‡* [ID]

1   Department of Mathematics, Faculty of Science, Damietta University, New Damietta 34517, Egypt
2   Faculty of Mathematics and Computer Science, Babeş-Bolyai University, 400084 Cluj-Napoca, Romania; bulboaca@math.ubbcluj.ro
*   Correspondence: shezaeldeeb@yahoo.com; Tel.: +966-0557689615
†   Current address: Department of Mathematics, College of Science and Arts in Al-Badaya, Al-Qassim University, Al-Badaya 3803070, Saudi Arabia.
‡   These authors contributed equally to this work.

**Abstract:** In this paper, we obtain some applications of the theory of differential subordination, differential superordination, and sandwich-type results for some subclasses of symmetric functions connected with a *q*-analog integral operator.

**Keywords:** symmetric functions; Hadamard (convolution) product; differential subordination; differential superordination; sandwich-type results; integral operator

## 1. Introduction

The theory of *q*-analysis has an important role in many areas of mathematics and physics. Jackson [1,2] was the first that gave some application of *q*-calculus and introduced the *q*-analog of derivative and integral operator (see also [3]). Let $\mathcal{H}(\mathbb{U})$ denote the class of analytic functions in the open unit disk $\mathbb{U} := \{z \in \mathbb{C} : |z| < 1\}$, and $\mathcal{H}[a, m]$ denote the subclass of functions $f \in \mathcal{H}(\mathbb{U})$ of the form

$$f(z) = a + a_m z^m + a_{m+1} z^{m+1} + \dots, z \in \mathbb{U},$$

with $a \in \mathbb{C}$ and $m \in \mathbb{N} := \{1, 2, \dots\}$.

In addition, let $\mathcal{A}(m)$ denote the subclass of functions $f \in \mathcal{H}(\mathbb{U})$ of the form

$$f(z) = z + \sum_{k=m+1}^{\infty} a_k z^k, z \in \mathbb{U}, \tag{1}$$

with $m \in \mathbb{N}$, and let $\mathcal{A} := \mathcal{A}(1)$.

We define the integral operator $\mathcal{K}_{n,m}^{\alpha} : \mathcal{A}(m) \to \mathcal{A}(m)$, with $\alpha > 0$ and $n \geq 0$, as follows:

$$\mathcal{K}_{n,m}^{0} f := f,$$

and

$$\mathcal{K}_{n,m}^{\alpha} f(z) := \frac{(n+1)^{\alpha}}{\Gamma(\alpha) z^n} \int_{0}^{z} t^{n-1} \left(\log \frac{z}{t}\right)^{\alpha-1} f(t) dt,$$

where all the powers are the principal ones, and $\log 1 = 0$.

If $f \in \mathcal{A}(m)$ has the power expansion of the form in Equation (1), it can be easily verified that

$$\mathcal{K}_{n,m}^{\alpha} f(z) = z + \sum_{k=m+1}^{\infty} \left(\frac{n+1}{n+k}\right)^{\alpha} a_k z^k, \ z \in \mathbb{U}.$$

For $0 < q < 1$, the $q$-derivative of the operator $\mathcal{K}_{n,m}^{\alpha}$ is defined by

$$\partial_q \mathcal{K}_{n,m}^{\alpha} f(z) := \frac{\mathcal{K}_{n,m}^{\alpha} f(qz) - \mathcal{K}_{n,m}^{\alpha} f(z)}{z(q-1)}, \ z \in \mathbb{U},$$

that is

$$\partial_q \left[ z + \sum_{k=m+1}^{\infty} \left(\frac{n+1}{n+k}\right)^{\alpha} a_k z^k \right] = 1 + \sum_{k=m+1}^{\infty} \left(\frac{n+1}{n+k}\right)^{\alpha} [k,q] a_k z^{k-1}, \ z \in \mathbb{U}, \tag{2}$$

where

$$[k,q] = \frac{1-q^k}{1-q} = 1 + \sum_{i=1}^{k-1} q^i, \quad [0,q] = 0,$$

It is easily to verify from Equation (2) that

$$z\partial_q \mathcal{K}_{n,m}^{\alpha} f(z) = z + \sum_{k=m+1}^{\infty} \left(\frac{n+1}{n+k}\right)^{\alpha} [k,q] a_k z^k, \ z \in \mathbb{U}.$$

For any non negative integer $k$, the $q$-number shift factorial is given by

$$[k,q]! = \begin{cases} 1, & \text{if} \quad k=0, \\ [1,q]\,[2,q]\,[3,q]\dots[k,q], & \text{if} \quad k \in \mathbb{N}, \end{cases}$$

while the $q$-generalized Pochhammer symbol for $r > 0$ is defined by

$$[r,q]_k = \begin{cases} 1, & \text{if} \quad k=0, \\ [r,q]\,[r+1,q]\dots[r+k-1,q], & \text{if} \quad k \in \mathbb{N}. \end{cases}$$

For $\lambda > -1$, we define the operator $\mathcal{N}_{n,m,q}^{\lambda,\alpha} : \mathcal{A}(m) \to \mathcal{A}(m)$ by

$$\mathcal{N}_{n,m,q}^{\lambda,\alpha} f(z) * \mathcal{M}_{q,\lambda+1}(z) = z\partial_q \mathcal{K}_{n,m}^{\alpha} f(z),$$

where

$$\mathcal{M}_{q,\lambda+1}(z) := z + \sum_{k=m+1}^{\infty} \frac{[\lambda+1,q]_{k-1}}{[k-1,q]!} z^k, \ z \in \mathbb{U}.$$

From the above definition, we obtain

$$\mathcal{N}_{n,m,q}^{\lambda,\alpha} f(z) = z + \sum_{k=m+1}^{\infty} \left(\frac{n+1}{n+k}\right)^{\alpha} \frac{[k,q]\,[k-1,q]!}{[\lambda+1,q]_{k-1}} a_k z^k$$

$$= z + \sum_{k=m+1}^{\infty} \frac{[k,q]!}{[\lambda+1,q]_{k-1}} \left(\frac{n+1}{n+k}\right)^{\alpha} a_k z^k, \ z \in \mathbb{U}, \tag{3}$$

$$(\alpha > 0, \ \lambda > -1, \ m \geq 0, \ 0 < q < 1)$$

and from Equation (3) we can easily verify that

$$[\lambda+1,q]\mathcal{N}_{n,m,q}^{\lambda,\alpha} f(z) = [\lambda,q]\,\mathcal{N}_{n,m,q}^{\lambda+1,\alpha} f(z) + q^{\lambda} z\partial_q \mathcal{N}_{n,m,q}^{\lambda+1,\alpha} f(z), \ z \in \mathbb{U}.$$

We note that

$$\lim_{q \to 1^-} \mathcal{N}_{n,m,q}^{\lambda,\alpha} f(z) =: \mathcal{I}_{n,m}^{\lambda,\alpha} f(z) = z + \sum_{k=m+1}^{\infty} \frac{k!}{(\lambda+1)_{k-1}} \left( \frac{n+1}{n+k} \right)^{\alpha} a_k z^k, \ z \in \mathbb{U}. \tag{4}$$

**Definition 1.** *For $f, g \in \mathcal{H}(\mathbb{U})$, we say that $f$ is subordinate to $g$, written $f(z) \prec g(z)$, if there exists a Schwarz function $w$, which is analytic in $\mathbb{U}$, with $w(0) = 0$ and $|w(z)| < 1$ for all $z \in \mathbb{U}$, such that $f(z) = g(w(z)), z \in \mathbb{U}$. Furthermore, if the function $g$ is univalent in $\mathbb{U}$, then we have the following equivalence (see [4,5]):*

$$f(z) \prec g(z) \Leftrightarrow f(0) = g(0) \text{ and } f(\mathbb{U}) \subset g(\mathbb{U}).$$

Let $k, h \in \mathcal{H}(\mathbb{U})$, and let $\varphi(r, s; z) : \mathbb{C}^2 \times \mathbb{U} \to \mathbb{C}$.
(i) If $k$ satisfies *the first order differential subordination*

$$\varphi(k(z), zk'(z); z) \prec h(z), \tag{5}$$

then $k$ is said to be *a solution of the differential subordination* in Equation (5). The function $q$ is called *a dominant of the solutions of the differential subordination* in Equation (5) if $k(z) \prec q(z)$ for all the functions $k$ satisfying Equation (5). A dominant $\widetilde{q}$ is said to be *the best dominant of* Equation (5) if $\widetilde{q}(z) \prec q(z)$ for all the dominants $q$.
(ii) If $k$ satisfies *the first order differential superordination*

$$h(z) \prec \varphi(k(z), zk'(z); z), \tag{6}$$

then $k$ is called to be *a solution of the differential superordination* in Equation (6). The function $q$ is called *a subordinant of the solutions of the differential superordination* in Equation (6) if $q(z) \prec k(z)$ for all the functions $k$ satisfying Equation (6). A subordinant $\widetilde{q}$ is said to be *the best subordinant of* Equation (6) if $q(z) \prec \widetilde{q}(z)$ for all the subordinants $q$.

Miller and Mocanu [6] obtained conditions on the functions $h$, $q$ and $\varphi$ for which the following implication holds:

$$h(z) \prec \varphi(k(z), zk'(z); z) \Rightarrow q(z) \prec k(z).$$

Applying these methods, in [7,8], the author studied general classes of first order differential superordinations and superordination-preserving integral operators. Using the results of Bulboacă [4] (see also [9,10]), the authors of [11] obtained sufficient conditions for functions $f \in \mathcal{A}$ to satisfy the double subordination

$$q_1(z) \prec \frac{zf'(z)}{f(z)} \prec q_2(z),$$

where $q_1$ and $q_2$ are univalent functions in $\mathbb{U}$, normalized with $q_1(0) = q_2(0) = 1$.

Sakaguchi [12] introduced a class $S_s^*$ of *functions starlike with respect to symmetric points*, which consists of functions $f \in \mathcal{A}$ satisfying the inequality

$$\operatorname{Re} \frac{zf'(z)}{f(z) - f(-z)} > 0, \ z \in \mathbb{U},$$

that represents a subclass of *close-to-convex functions*, and hence univalent in $\mathbb{U}$. Moreover, this class includes the class of *convex functions* and *odd starlike functions with respect to the origin* (see [12,13]).

In addition, Aouf et al. [14] introduced and studied the class $S_{s,n}^* T(1,1)$ of *functions n-starlike with respect to symmetric points*, which consists of functions $f \in \mathcal{A}$, with $a_k \leq 0$ for $k \geq 2$, and satisfying the inequality

$$\operatorname{Re} \frac{D^{n+1} f(z)}{D^n f(z) - D^n f(-z)} > 0, \ z \in \mathbb{U},$$

where $D^n$ is the *Sălăgean operator* [15].

The classes defined in [12,13] could be generalized by introducing the next class of functions, defined with the aid of the $\mathcal{N}_{n,m,q}^{\lambda,\alpha}$ operator defined as follows:

**Definition 2.** *A function $f \in \mathcal{A}(m)$ with*

$$\mathcal{N}_{n,m,q}^{\lambda,\alpha}f(z) - \mathcal{N}_{n,m,q}^{\lambda,\alpha}f(-z) \neq 0, \ z \in \dot{\mathbb{U}} := \mathbb{U} \setminus \{0\}, \tag{7}$$

*is said to be in the class $\mathcal{M}_{n,m,q}^{\lambda,\alpha}(\gamma, \mu, A, B)$ if it satisfies the subordination condition*

$$(1+\gamma) \left( \frac{2z}{\mathcal{N}_{n,m,q}^{\lambda,\alpha}f(z) - \mathcal{N}_{n,m,q}^{\lambda,\alpha}f(-z)} \right)^{\mu}$$
$$-\gamma \left( \frac{z\left(\mathcal{N}_{n,m,q}^{\lambda,\alpha}f(z)\right)' - z\left(\mathcal{N}_{n,m,q}^{\lambda,\alpha}f(-z)\right)'}{\mathcal{N}_{n,m,q}^{\lambda,\alpha}f(z) - \mathcal{N}_{n,m,q}^{\lambda,\alpha}f(-z)} \right) \left( \frac{2z}{\mathcal{N}_{n,m,q}^{\lambda,\alpha}f(z) - \mathcal{N}_{n,m,q}^{\lambda,\alpha}f(-z)} \right)^{\mu} \prec \frac{1+Az}{1+Bz}, \tag{8}$$
$$(\gamma \in \mathbb{C}, \ 0 < \mu < 1, \ -1 \leq B < A \leq 1, \ m \in \mathbb{N}, \ \alpha > 0, \ n \geq 0, \ 0 < q < 1, \ \lambda > -1).$$

By specializing the parameters $\alpha$, $\lambda$ and $q$, we obtain the following subclasses:
(i) For $q \to 1^-$, we define the class $\mathcal{W}_{n,m}^{\lambda,\alpha}(\gamma, \mu, A, B)$ as follows:

$$\mathcal{W}_{n,m}^{\lambda,\alpha}(\gamma, \mu, A, B) := \left\{ f \in \mathcal{A}(m) : (1+\gamma) \left( \frac{2z}{\mathcal{I}_{n,m}^{\lambda,\alpha}f(z) - \mathcal{I}_{n,m}^{\lambda,\alpha}f(-z)} \right)^{\mu} \right.$$
$$\left. -\gamma \left( \frac{z\left(\mathcal{I}_{n,m}^{\lambda,\alpha}f(z)\right)' - z\left(\mathcal{I}_{n,m}^{\lambda,\alpha}f(-z)\right)'}{\mathcal{I}_{n,m}^{\lambda,\alpha}f(z) - \mathcal{I}_{n,m}^{\lambda,\alpha}f(-z)} \right) \left( \frac{2z}{\mathcal{I}_{n,m}^{\lambda,\alpha}f(z) - \mathcal{I}_{n,m}^{\lambda,\alpha}f(-z)} \right)^{\mu} \prec \frac{1+Az}{1+Bz} \right\},$$

where the operator $\mathcal{I}_{n,m}^{\lambda,\alpha}$ is defined by Equation (4);
(ii) For $q \to 1^-$, $\alpha = 0$ and $\lambda = 1$, we define the class $\mathcal{N}^{\gamma,\mu}(m, A, B)$ that corrects the class defined by Muhammad and Marwan [16] as follows:

$$\mathcal{N}^{\gamma,\mu}(m, A, B) := \left\{ f \in \mathcal{A}(m) : (1+\gamma) \left( \frac{2z}{f(z) - f(-z)} \right)^{\mu} \right.$$
$$\left. -\gamma \left( \frac{z\left(f'(z) - f'(-z)\right)}{f(z) - f(-z)} \right) \left( \frac{2z}{f(z) - f(-z)} \right)^{\mu} \prec \frac{1+Az}{1+Bz} \right\}.$$

In this paper, we obtain some sharp differential subordination and superordination results for the functions belonging to the class $\mathcal{M}_{n,m,q}^{\lambda,\alpha}(\gamma, \mu, A, B)$ to try to make a connection between a special subclass of analytic functions whose coefficients are given by the $q$-analog of integral operator and the differential subordination theory.

## 2. Preliminaries

To prove our results, we need the following definition and lemmas.

**Definition 3** ([5])**.** *(Definition 2.2b., p. 21) Let $\mathcal{Q}$ be the set of all functions $f$ that are analytic and injective on $\overline{\mathbb{U}} \setminus E(f)$, where $E(f) := \left\{ \zeta \in \partial\mathbb{U} : \lim_{z \to \zeta} f(z) = \infty \right\}$ and are such that $f'(\zeta) \neq 0$ for $\zeta \in \partial\mathbb{U} \setminus E(f)$.*

**Lemma 1** ([5])**.** *(Theorem 3.1b., p. 71) Let the function $H$ be convex in $\mathbb{U}$, with $H(0) = a$, and $\zeta \neq 0$ with $\operatorname{Re}\zeta \geq 0$. If $\Phi \in \mathcal{H}[a, m]$ and*

$$\Phi(z) + \frac{z\Phi'(z)}{\zeta} \prec H(z), \tag{9}$$

*then*

$$\Phi(z) \prec \Psi(z) := \frac{\zeta}{mz^{\frac{\zeta}{m}}} \int_0^z t^{\frac{\zeta}{m}-1} H(t)dt \prec H(z),$$

*and the function $\Psi$ is convex, $\Psi \in \mathcal{H}[a, m]$, and is the best dominant of Equation (9).*

**Lemma 2** ([17])**.** *(Lemma 2.2., p. 3) Let $q$ be univalent in $\mathbb{U}$, with $q(0) = 1$. Let $\xi, \varphi \in \mathbb{C}$ with $\varphi \neq 0$, and assume that*

$$\text{Re}\left(1 + \frac{zq''(z)}{q'(z)}\right) > \max\left\{0; -\text{Re}\frac{\xi}{\varphi}\right\}, \ z \in \mathbb{U}.$$

*If $k$ is analytic in $\mathbb{U}$ and*

$$\xi k(z) + \varphi z k'(z) \prec \xi q(z) + \varphi z q'(z), \tag{10}$$

*then $k(z) \prec q(z)$, and $q$ is the best dominant of Equation (10).*

From [6] (Theorem 6, p. 820), we could easily obtain the following lemma:

**Lemma 3.** *Let $q$ be convex in $\mathbb{U}$, and $k \neq 0$ with $\text{Re } k \geq 0$. If $g \in \mathcal{H}[q(0), 1] \cap \mathcal{Q}$, such that $g(z) + kzg'(z)$ is univalent in $\mathbb{U}$, then*

$$q(z) + kzq'(z) \prec g(z) + kzg'(z), \tag{11}$$

*implies that $q(z) \prec g(z)$, and $q$ is the best subordinant of Equation (11).*

**Lemma 4** ([18])**.** *Let $F$ be analytic and convex in $\mathbb{U}$, and $0 \leq \lambda \leq 1$. If $f, g \in \mathcal{A}$, such that $f(z) \prec F(z)$ and $g(z) \prec F(z)$, then*

$$\lambda f(z) + (1-\lambda)g(z) \prec F(z).$$

## 3. Main Results

Unless otherwise mentioned, we assume in the remainder of this paper that $\gamma \in \mathbb{C}, 0 < \mu < 1$, $-1 \leq B < A \leq 1, m \in \mathbb{N}, \alpha > 0, n \geq 0, 0 < q < 1, \lambda > -1$, and all the powers are understood as principle values.

**Theorem 1.** *If $f \in \mathcal{M}_{n,m,q}^{\lambda,\alpha}(\gamma, \mu, A, B)$ and $\gamma \in \mathbb{C}^* := \mathbb{C} \setminus \{0\}$ with $\text{Re } \gamma \geq 0$, then*

$$\left(\frac{2z}{\mathcal{N}_{n,m,q}^{\lambda,\alpha}f(z) - \mathcal{N}_{n,m,q}^{\lambda,\alpha}f(-z)}\right)^\mu \prec \Psi(z) := \frac{\mu}{\gamma m} \int_0^1 \frac{1 + Azu}{1 + Bzu} u^{\frac{\mu}{\gamma m}-1} du \prec \frac{1 + Az}{1 + Bz},$$

*and $\Psi$ is convex, $\Psi \in \mathcal{H}[1, m]$, and is the best dominant.*

**Proof.** If we define the function $h$ by

$$h(z) := \left(\frac{2z}{\mathcal{N}_{n,m,q}^{\lambda,\alpha}f(z) - \mathcal{N}_{n,m,q}^{\lambda,\alpha}f(-z)}\right)^\mu, \ z \in \mathbb{U}, \tag{12}$$

from Equation (7), it follows that $h$ is an analytic function in $\mathbb{U}$, with $h(0) = 1$. Differentiating Equation (12) with respect to $z$, we obtain that

$$(1+\gamma)\left(\frac{2z}{\mathcal{N}_{n,m,q}^{\lambda,\alpha}f(z)-\mathcal{N}_{n,m,q}^{\lambda,\alpha}f(-z)}\right)^{\mu}$$

$$-\gamma\left(\frac{z\left(\mathcal{N}_{n,m,q}^{\lambda,\alpha}f(z)\right)'-z\left(\mathcal{N}_{n,m,q}^{\lambda,\alpha}f(-z)\right)'}{\mathcal{N}_{n,m,q}^{\lambda,\alpha}f(z)-\mathcal{N}_{n,m,q}^{\lambda,\alpha}f(-z)}\right)\left(\frac{2z}{\mathcal{N}_{n,m,q}^{\lambda,\alpha}f(z)-\mathcal{N}_{n,m,q}^{\lambda,\alpha}f(-z)}\right)^{\mu} \qquad (13)$$

$$= h(z)+\frac{\gamma}{\mu}zh'(z) \prec \frac{1+Az}{1+Bz}.$$

Since

$$\mathcal{N}_{n,m,q}^{\lambda,\alpha}f(z)=z+\sum_{k=m+1}^{\infty}\alpha_k z^k, \quad \text{and} \quad \mathcal{N}_{n,m,q}^{\lambda,\alpha}f(-z)=-z+\sum_{k=m+1}^{\infty}\alpha_k(-1)^k z^k,$$

where

$$\alpha_k=\frac{[k,q]!}{[\lambda+1,q]_{k-1}}\left(\frac{n+1}{n+k}\right)^{\alpha}a_k, \; k\geq m+1,$$

we have

$$U(z):=\frac{2z}{\mathcal{N}_{n,m,q}^{\lambda,\alpha}f(z)-\mathcal{N}_{n,m,q}^{\lambda,\alpha}f(-z)}=\frac{2z}{2z+\sum\limits_{k=m+1}^{\infty}\alpha_k\left[1+(-1)^{k+1}\right]z^k}=\frac{1}{1+\sum\limits_{s=m}^{\infty}\beta_s z^s},$$

with

$$\beta_s=\frac{\alpha_{s+1}\left[1+(-1)^s\right]}{2}, \; s\geq m.$$

Moreover,

$$U(z)=\frac{1}{1+\sum\limits_{s=m}^{\infty}\beta_s z^s}=1+\sum_{j=1}^{\infty}\gamma_j z^j, \; z\in\mathbb{U},$$

with unknowns $\gamma_j$, $j\geq 1$, we have

$$1=\left(1+\beta_m z^m+\beta_{m+1}z^{m+1}+\dots\right)\left(1+\gamma_1 z+\gamma_2 z^2+\dots+\gamma_m z^m+\gamma_{m+1}z^{m+1}+\dots\right),$$

and equating the corresponding coefficients it follows that

$$\gamma_1=\gamma_2=\dots=\gamma_{m-1}=0, \quad \gamma_m=-\beta_m, \quad \gamma_{m+1}=-\beta_{m+1},\dots,$$

hence

$$U(z)=1+\sum_{j=m}^{\infty}\gamma_j z^j\in\mathcal{H}[1,m].$$

According to Equation (12), we have

$$h=U^{\mu}, \quad \text{with} \quad U\in\mathcal{H}[1,m],$$

and using the binomial power expansion formula, we get

$$h=U^{\mu}\in\mathcal{H}[1,m].$$

Now, from the subordination in Equation (13), using Lemma 1 for $\zeta=\dfrac{\mu}{\gamma}$, we obtain our result. $\square$

Taking $q\to 1^-$ in Theorem 1, we obtain the following corollary:

**Corollary 1.** *If $f \in \mathcal{W}_{n,m}^{\lambda,\alpha}(\gamma,\mu,A,B)$ and $\gamma \in \mathbb{C}^* := \mathbb{C} \setminus \{0\}$ with $\operatorname{Re} \gamma \geq 0$, then*

$$\left( \frac{2z}{\mathcal{I}_{n,m}^{\lambda,\alpha}f(z) - \mathcal{I}_{n,m}^{\lambda,\alpha}f(-z)} \right)^{\mu} \prec \Psi(z) := \frac{\mu}{\gamma m} \int_0^1 \frac{1 + Azu}{1 + Bzu} u^{\frac{\mu}{\gamma m} - 1} du \prec \frac{1 + Az}{1 + Bz},$$

*and $\Psi$ is convex, $\Psi \in \mathcal{H}[1, m]$, and is the best dominant.*

**Remark 1.** *The above theorem shows that*

$$\mathcal{M}_{n,m,q}^{\lambda,\alpha}(\gamma,\mu,A,B) \subset \mathcal{M}_{n,m,q}^{\lambda,\alpha}(0,\mu,A,B),$$

*for all $\gamma \in \mathbb{C}$ with $\operatorname{Re} \gamma \geq 0$.*

Moreover, the next inclusion result for the classes $\mathcal{M}_{n,m,q}^{\lambda,\alpha}(\gamma,\mu,A,B)$ holds:

**Theorem 2.** *If $\gamma_1, \gamma_2 \in \mathbb{R}$ such that $0 \leq \gamma_1 \leq \gamma_2$, and $-1 \leq B_1 \leq B_2 < A_2 \leq A_1 \leq 1$, then*

$$\mathcal{M}_{n,m,q}^{\lambda,\alpha}(\gamma_2,\mu,A_2,B_2) \subset \mathcal{M}_{n,m,q}^{\lambda,\alpha}(\gamma_1,\mu,A_1,B_1). \tag{14}$$

**Proof.** If $f \in \mathcal{M}_{n,m,q}^{\lambda,\alpha}(\gamma_2,\mu,A_2,B_2)$, since $-1 \leq B_1 \leq B_2 < A_2 \leq A_1 \leq 1$, it is easy to check that

$$\begin{aligned}
(1 + \gamma_2) & \left( \frac{2z}{\mathcal{N}_{n,m,q}^{\lambda,\alpha}f(z) - \mathcal{N}_{n,m,q}^{\lambda,\alpha}f(-z)} \right)^{\mu} \\
- \gamma_2 & \left( \frac{z\left( \mathcal{N}_{n,m,q}^{\lambda,\alpha}f(z) - \mathcal{N}_{n,m,q}^{\lambda,\alpha}f(-z) \right)'}{\mathcal{N}_{n,m,q}^{\lambda,\alpha}f(z) - \mathcal{N}_{n,m,q}^{\lambda,\alpha}f(-z)} \right) \left( \frac{2z}{\mathcal{N}_{n,m,q}^{\lambda,\alpha}f(z) - \mathcal{N}_{n,m,q}^{\lambda,\alpha}f(-z)} \right)^{\mu} \\
& \prec \frac{1 + A_2 z}{1 + B_2 z} \prec \frac{1 + A_1 z}{1 + B_1 z},
\end{aligned} \tag{15}$$

that is $f \in \mathcal{M}_{n,m,q}^{\lambda,\alpha}(\gamma_1,\mu,A_1,B_1)$, hence the assertion in Equation (14) holds for $\gamma_1 = \gamma_2$.

If $0 \leq \gamma_1 < \gamma_2$, from Remark 1 and Equation (15), it follows $f \in \mathcal{M}_{n,m,q}^{\lambda,\alpha}(0,\mu,A_1,B_1)$, that is

$$\left( \frac{2z}{\mathcal{N}_{n,m,q}^{\lambda,\alpha}f(z) - \mathcal{N}_{n,m,q}^{\lambda,\alpha}f(-z)} \right)^{\mu} \prec \frac{1 + A_1 z}{1 + B_1 z}. \tag{16}$$

A simple computation shows that

$$\begin{aligned}
(1 + \gamma_1) & \left( \frac{2z}{\mathcal{N}_{n,m,q}^{\lambda,\alpha}f(z) - \mathcal{N}_{n,m,q}^{\lambda,\alpha}f(-z)} \right)^{\mu} \\
- \gamma_1 & \left( \frac{z\left( \mathcal{N}_{n,m,q}^{\lambda,\alpha}f(z) - \mathcal{N}_{n,m,q}^{\lambda,\alpha}f(-z) \right)'}{\mathcal{N}_{n,m,q}^{\lambda,\alpha}f(z) - \mathcal{N}_{n,m,q}^{\lambda,\alpha}f(-z)} \right) \left( \frac{2z}{\mathcal{N}_{n,m,q}^{\lambda,\alpha}f(z) - \mathcal{N}_{n,m,q}^{\lambda,\alpha}f(-z)} \right)^{\mu} \\
= & \left( 1 - \frac{\gamma_1}{\gamma_2} \right) \left( \frac{2z}{\mathcal{N}_{n,m,q}^{\lambda,\alpha}f(z) - \mathcal{N}_{n,m,q}^{\lambda,\alpha}f(-z)} \right)^{\mu} \\
& + \frac{\gamma_1}{\gamma_2} \left[ (1 + \gamma_2) \left( \frac{2z}{\mathcal{N}_{n,m,q}^{\lambda,\alpha}f(z) - \mathcal{N}_{n,m,q}^{\lambda,\alpha}f(-z)} \right)^{\mu} \right. \\
& \left. - \gamma_2 \left( \frac{z\left( \mathcal{N}_{n,m,q}^{\lambda,\alpha}f(z) - \mathcal{N}_{n,m,q}^{\lambda,\alpha}f(-z) \right)'}{\mathcal{N}_{n,m,q}^{\lambda,\alpha}f(z) - \mathcal{N}_{n,m,q}^{\lambda,\alpha}f(-z)} \right) \left( \frac{2z}{\mathcal{N}_{n,m,q}^{\lambda,\alpha}f(z) - \mathcal{N}_{n,m,q}^{\lambda,\alpha}f(-z)} \right)^{\mu} \right], \quad z \in \mathbb{U}.
\end{aligned} \tag{17}$$

Moreover,

$$0 \leq \frac{\gamma_1}{\gamma_2} < 1,$$

and the function $\dfrac{1 + A_1 z}{1 + B_1 z}$, with $-1 \leq B_1 < A_1 \leq 1$, is analytic and convex in $\mathbb{U}$. According to Equation (17), using the subordinations in Equations (15) and (16), from Lemma 4, we deduce that

$$(1 + \gamma_1) \left( \frac{2z}{\mathcal{N}_{n,m,q}^{\lambda,\alpha} f(z) - \mathcal{N}_{n,m,q}^{\lambda,\alpha} f(-z)} \right)^{\mu}$$

$$-\gamma_1 \left( \frac{z \left( \mathcal{N}_{n,m,q}^{\lambda,\alpha} f(z) - \mathcal{N}_{n,m,q}^{\lambda,\alpha} f(-z) \right)'}{\mathcal{N}_{n,m,q}^{\lambda,\alpha} f(z) - \mathcal{N}_{n,m,q}^{\lambda,\alpha} f(-z)} \right) \left( \frac{2z}{\mathcal{N}_{n,m,q}^{\lambda,\alpha} f(z) - \mathcal{N}_{n,m,q}^{\lambda,\alpha} f(-z)} \right)^{\mu} \prec \frac{1 + A_1 z}{1 + B_1 z},$$

that is $f \in \mathcal{M}_{n,m,q}^{\lambda,\alpha}(\gamma_1, \mu, A_1, B_1)$. □

Taking $q \to 1^-$ in Theorem 2, we obtain the following corollary:

**Corollary 2.** *If $\gamma_1, \gamma_2 \in \mathbb{R}$ such that $0 \leq \gamma_1 \leq \gamma_2$, and $-1 \leq B_1 \leq B_2 < A_2 \leq A_1 \leq 1$, then*

$$\mathcal{W}_{n,m}^{\lambda,\alpha}(\gamma_2, \mu, A_2, B_2) \subset \mathcal{W}_{n,m}^{\lambda,\alpha}(\gamma_1, \mu, A_1, B_1).$$

**Example 1.** *For the special case $A_1 = 1$ and $B_1 = -1$, Theorem 2 and Corollary 2 reduce to the next examples, respectively:*
*Suppose that $\gamma_1, \gamma_2 \in \mathbb{R}$ such that $0 \leq \gamma_1 \leq \gamma_2$, and $-1 \leq B_2 < A_2 \leq 1$.*
*1. If $f \in \mathcal{M}_{n,m,q}^{\lambda,\alpha}(\gamma_2, \mu, A_2, B_2)$, then*

$$\mathrm{Re} \left\{ (1 + \gamma_1) \left( \frac{2z}{\mathcal{N}_{n,m,q}^{\lambda,\alpha} f(z) - \mathcal{N}_{n,m,q}^{\lambda,\alpha} f(-z)} \right)^{\mu} \right.$$

$$\left. -\gamma_1 \left( \frac{z \left( \mathcal{N}_{n,m,q}^{\lambda,\alpha} f(z) - \mathcal{N}_{n,m,q}^{\lambda,\alpha} f(-z) \right)'}{\mathcal{N}_{n,m,q}^{\lambda,\alpha} f(z) - \mathcal{N}_{n,m,q}^{\lambda,\alpha} f(-z)} \right) \left( \frac{2z}{\mathcal{N}_{n,m,q}^{\lambda,\alpha} f(z) - \mathcal{N}_{n,m,q}^{\lambda,\alpha} f(-z)} \right)^{\mu} \right\} > 0, \ z \in \mathbb{U};$$

*2. If $f \in \mathcal{W}_{n,m}^{\lambda,\alpha}(\gamma_2, \mu, A_2, B_2)$, then*

$$\mathrm{Re} \left\{ (1 + \gamma_1) \left( \frac{2z}{\mathcal{I}_{n,m,q}^{\lambda,\alpha} f(z) - \mathcal{I}_{n,m,q}^{\lambda,\alpha} f(-z)} \right)^{\mu} \right.$$

$$\left. -\gamma_1 \left( \frac{z \left( \mathcal{I}_{n,m,q}^{\lambda,\alpha} f(z) - \mathcal{I}_{n,m,q}^{\lambda,\alpha} f(-z) \right)'}{\mathcal{I}_{n,m,q}^{\lambda,\alpha} f(z) - \mathcal{I}_{n,m,q}^{\lambda,\alpha} f(-z)} \right) \left( \frac{2z}{\mathcal{I}_{n,m,q}^{\lambda,\alpha} f(z) - \mathcal{I}_{n,m,q}^{\lambda,\alpha} f(-z)} \right)^{\mu} \right\} > 0, \ z \in \mathbb{U};$$

**Theorem 3.** *Suppose that $q$ is univalent in $\mathbb{U}$, with $q(0) = 1$, and let $\gamma \in \mathbb{C}^*$ such that*

$$\mathrm{Re} \left( 1 + \frac{zq''(z)}{q'(z)} \right) > \max \left\{ 0; -\mathrm{Re}\,\frac{\mu}{\gamma} \right\}, \ z \in \mathbb{U}. \tag{18}$$

*If $f \in \mathcal{A}(m)$ such that Equation (7) holds, and satisfies the subordination*

$$(1 + \gamma) \left( \frac{2z}{\mathcal{N}_{n,m,q}^{\lambda,\alpha} f(z) - \mathcal{N}_{n,m,q}^{\lambda,\alpha} f(-z)} \right)^{\mu}$$

$$-\gamma \left( \frac{z \left( \mathcal{N}_{n,m,q}^{\lambda,\alpha} f(z) - \mathcal{N}_{n,m,q}^{\lambda,\alpha} f(-z) \right)'}{\mathcal{N}_{n,m,q}^{\lambda,\alpha} f(z) - \mathcal{N}_{n,m,q}^{\lambda,\alpha} f(-z)} \right) \left( \frac{2z}{\mathcal{N}_{n,m,q}^{\lambda,\alpha} f(z) - \mathcal{N}_{n,m,q}^{\lambda,\alpha} f(-z)} \right)^{\mu} \tag{19}$$

$$\prec q(z) + \frac{\gamma}{\mu} z q'(z),$$

*then*

$$\left(\frac{2z}{\mathcal{N}_{n,m,q}^{\lambda,\alpha}f(z) - \mathcal{N}_{n,m,q}^{\lambda,\alpha}f(-z)}\right)^{\mu} \prec q(z),$$

*and q is the best dominant of Equation* (19)*.*

**Proof.** Since $f \in \mathcal{A}(m)$ such that Equation (7) holds, it follows that the function $h$ defined by Equation (12) is analytic in $\mathbb{U}$, and $h(0) = 1$. As in the proof of Theorem 1, differentiating Equation (12) with respect to $z$, we obtain that Equation (19) is equivalent to

$$h(z) + \frac{\gamma}{\mu}zh'(z) \prec q(z) + \frac{\gamma}{\mu}zq'(z).$$

Using Lemma 2 for $\xi := 1$ and $\varphi := \dfrac{\gamma}{\mu}$, we get that the above subordination implies $h(z) \prec q(z)$, and $q$ is the best dominant of Equation (19). □

For the special case $q(z) = \dfrac{1 + Az}{1 + Bz}$, with $-1 \leq B < A \leq 1$, Theorem 3 reduces to the following corollary:

**Corollary 3.** *Let $\gamma \in \mathbb{C}^*$ and $-1 \leq B < A \leq 1$, such that*

$$\max\left\{-1; -\frac{1 + \operatorname{Re}\frac{\mu}{\gamma}}{1 - \operatorname{Re}\frac{\mu}{\gamma}}\right\} \leq B \leq 0, \quad or \quad 0 \leq B \leq \min\left\{1; \frac{1 + \operatorname{Re}\frac{\mu}{\gamma}}{1 - \operatorname{Re}\frac{\mu}{\gamma}}\right\}. \tag{20}$$

*If $f \in \mathcal{A}(m)$ such that Equation (7) holds, and satisfies the subordination*

$$(1 + \gamma)\left(\frac{2z}{\mathcal{N}_{n,m,q}^{\lambda,\alpha}f(z) - \mathcal{N}_{n,m,q}^{\lambda,\alpha}f(-z)}\right)^{\mu}$$
$$-\gamma\left(\frac{z\left(\mathcal{N}_{n,m,q}^{\lambda,\alpha}f(z) - \mathcal{N}_{n,m,q}^{\lambda,\alpha}f(-z)\right)'}{\mathcal{N}_{n,m,q}^{\lambda,\alpha}f(z) - \mathcal{N}_{n,m,q}^{\lambda,\alpha}f(-z)}\right)\left(\frac{2z}{\mathcal{N}_{n,m,q}^{\lambda,\alpha}f(z) - \mathcal{N}_{n,m,q}^{\lambda,\alpha}f(-z)}\right)^{\mu} \tag{21}$$
$$\prec \frac{1 + Az}{1 + Bz} + \frac{\gamma}{\mu}\frac{(A - B)z}{(1 + Bz)^2},$$

*then*

$$\left(\frac{2z}{\mathcal{N}_{n,m,q}^{\lambda,\alpha}f(z) - \mathcal{N}_{n,m,q}^{\lambda,\alpha}f(-z)}\right)^{\mu} \prec \frac{1 + Az}{1 + Bz},$$

*and $\dfrac{1 + Az}{1 + Bz}$ is the best dominant of Equation* (21)*.*

**Proof.** For $q(z) = \dfrac{1 + Az}{1 + Bz}$, the condition in Equation (18) reduces to

$$\operatorname{Re}\frac{1 - Bz}{1 + Bz} > \max\left\{0; -\operatorname{Re}\frac{\mu}{\gamma}\right\}, \quad z \in \mathbb{U}. \tag{22}$$

Since

$$\inf\left\{\operatorname{Re}\frac{1 - Bz}{1 + Bz} : z \in \mathbb{U}\right\} = \begin{cases} \dfrac{1 + B}{1 - B}, & \text{if } -1 \leq B \leq 0, \\[2mm] \dfrac{1 - B}{1 + B}, & \text{if } 0 \leq B < 1, \end{cases}$$

we easily check that Equation (22) holds if and only if the assumption in Equation (20) is satisfied, whenever $-1 \leq B < 1$. □

Taking $q \to 1^-$ in Theorem 3, we obtain the following corollary:

**Corollary 4.** *Suppose that $q$ is univalent in $\mathbb{U}$, with $q(0) = 1$, and let $\gamma \in \mathbb{C}^*$ such that*

$$\mathrm{Re}\left(1 + \frac{zq''(z)}{q'(z)}\right) > \max\left\{0; -\,\mathrm{Re}\,\frac{\mu}{\gamma}\right\}, \ z \in \mathbb{U}.$$

*If $f \in \mathcal{A}(m)$ such that Equation (7) holds, and satisfies the subordination*

$$(1 + \gamma)\left(\frac{2z}{\mathcal{I}_{n,m}^{\lambda,\alpha}f(z) - \mathcal{I}_{n,m}^{\lambda,\alpha}f(-z)}\right)^{\mu}$$

$$-\gamma\left(\frac{z\left(\mathcal{I}_{n,m}^{\lambda,\alpha}f(z) - \mathcal{I}_{n,m}^{\lambda,\alpha}f(-z)\right)'}{\mathcal{I}_{n,m}^{\lambda,\alpha}f(z) - \mathcal{I}_{n,m}^{\lambda,\alpha}f(-z)}\right)\left(\frac{2z}{\mathcal{I}_{n,m}^{\lambda,\alpha}f(z) - \mathcal{I}_{n,m}^{\lambda,\alpha}f(-z)}\right)^{\mu} \prec q(z) + \frac{\gamma}{\mu}zq'(z),$$

*then*

$$\left(\frac{2z}{\mathcal{I}_{n,m}^{\lambda,\alpha}f(z) - \mathcal{I}_{n,m}^{\lambda,\alpha}f(-z)}\right)^{\mu} \prec q(z),$$

*and $q$ is the best dominant of Equation (19).*

**Theorem 4.** *Let $q$ be convex in $\mathbb{U}$, with $q(0) = 1$, and $\gamma \in \mathbb{C}^*$, with $\mathrm{Re}\,\gamma \geq 0$. In addition, let $f \in \mathcal{A}(m)$ such that*

$$\left(\frac{2z}{\mathcal{N}_{n,m,q}^{\lambda,\alpha}f(z) - \mathcal{N}_{n,m,q}^{\lambda,\alpha}f(-z)}\right)^{\mu} \in \mathcal{H}[q(0), 1] \cap \mathcal{Q}, \tag{23}$$

*and assume that the function*

$$(1 + \gamma)\left(\frac{2z}{\mathcal{N}_{n,m,q}^{\lambda,\alpha}f(z) - \mathcal{N}_{n,m,q}^{\lambda,\alpha}f(-z)}\right)^{\mu}$$

$$-\gamma\left(\frac{z\left(\mathcal{N}_{n,m,q}^{\lambda,\alpha}f(z) - \mathcal{N}_{n,m,q}^{\lambda,\alpha}f(-z)\right)'}{\mathcal{N}_{n,m,q}^{\lambda,\alpha}f(z) - \mathcal{N}_{n,m,q}^{\lambda,\alpha}f(-z)}\right)\left(\frac{2z}{\mathcal{N}_{n,m,q}^{\lambda,\alpha}f(z) - \mathcal{N}_{n,m,q}^{\lambda,\alpha}f(-z)}\right)^{\mu} \tag{24}$$

*is univalent in $\mathbb{U}$.*

*If*

$$q(z) + \frac{\gamma}{\mu}zq'(z) \prec (1 + \gamma)\left(\frac{2z}{\mathcal{N}_{n,m,q}^{\lambda,\alpha}f(z) - \mathcal{N}_{n,m,q}^{\lambda,\alpha}f(-z)}\right)^{\mu}$$

$$-\gamma\left(\frac{z\left(\mathcal{N}_{n,m,q}^{\lambda,\alpha}f(z) - \mathcal{N}_{n,m,q}^{\lambda,\alpha}f(-z)\right)'}{\mathcal{N}_{n,m,q}^{\lambda,\alpha}f(z) - \mathcal{N}_{n,m,q}^{\lambda,\alpha}f(-z)}\right)\left(\frac{2z}{\mathcal{N}_{n,m,q}^{\lambda,\alpha}f(z) - \mathcal{N}_{n,m,q}^{\lambda,\alpha}f(-z)}\right)^{\mu}, \tag{25}$$

*then*

$$q(z) \prec \left(\frac{2z}{\mathcal{N}_{n,m,q}^{\lambda,\alpha}f(z) - \mathcal{N}_{n,m,q}^{\lambda,\alpha}f(-z)}\right)^{\mu},$$

*and $q$ is the best subordinant of Equation (25).*

**Proof.** Letting the function $h$ be defined by Equation (12), then $h \in \mathcal{H}[q(0), m]$, and from Equation (23) we have that $h \in \mathcal{H}[q(0), 1] \cap \mathcal{Q}$. As in the proof of Theorem 1, differentiating Equation (12) with respect to $z$, we obtain that

$$q(z) + \frac{\gamma}{\mu}zq'(z) \prec h(z) + \frac{\gamma}{\mu}zh'(z).$$

Now, according to Lemma 3 for $k := \dfrac{\gamma}{\mu}$ we obtain the desired result. $\square$

Taking $q(z) = \dfrac{1 + Az}{1 + Bz}$, with $-1 \leq B < A \leq 1$, in Theorem 4, we obtain the following corollary:

**Corollary 5.** *Let $\gamma \in \mathbb{C}^*$, with $\operatorname{Re}\gamma \geq 0$, and $-1 \leq B < A \leq 1$. If $f \in \mathcal{A}(m)$ such that the assumptions in Equations (23) and (24) hold, and satisfies the subordination*

$$\frac{1+Az}{1+Bz} + \frac{\gamma}{\mu}\frac{(A-B)z}{(1+Bz)^2} \prec (1+\gamma)\left(\frac{2z}{\mathcal{N}_{n,m,q}^{\lambda,\alpha}f(z) - \mathcal{N}_{n,m,q}^{\lambda,\alpha}f(-z)}\right)^{\mu}$$

$$-\gamma \left(\frac{z\left(\mathcal{N}_{n,m,q}^{\lambda,\alpha}f(z) - \mathcal{N}_{n,m,q}^{\lambda,\alpha}f(-z)\right)'}{\mathcal{N}_{n,m,q}^{\lambda,\alpha}f(z) - \mathcal{N}_{n,m,q}^{\lambda,\alpha}f(-z)}\right)\left(\frac{2z}{\mathcal{N}_{n,m,q}^{\lambda,\alpha}f(z) - \mathcal{N}_{n,m,q}^{\lambda,\alpha}f(-z)}\right)^{\mu}, \qquad (26)$$

*then*

$$\frac{1+Az}{1+Bz} \prec \left(\frac{2z}{\mathcal{N}_{n,m,q}^{\lambda,\alpha}f(z) - \mathcal{N}_{n,m,q}^{\lambda,\alpha}f(-z)}\right)^{\mu},$$

*and $\dfrac{1+Az}{1+Bz}$ is the best subordinant of Equation (26).*

Taking $q \to 1^-$ in Theorem 4, we obtain the following corollary:

**Corollary 6.** *Let $q$ be convex in $\mathbb{U}$, with $q(0) = 1$, and $\gamma \in \mathbb{C}^*$, with $\operatorname{Re}\gamma \geq 0$. In addition, let $f \in \mathcal{A}(m)$ such that*

$$\left(\frac{2z}{\mathcal{I}_{n,m}^{\lambda,\alpha}f(z) - \mathcal{I}_{n,m}^{\lambda,\alpha}f(-z)}\right)^{\mu} \in \mathcal{H}[q(0),1] \cap \mathcal{Q},$$

*and assume that the function*

$$(1+\gamma)\left(\frac{2z}{\mathcal{I}_{n,m}^{\lambda,\alpha}f(z) - \mathcal{I}_{n,m}^{\lambda,\alpha}f(-z)}\right)^{\mu}$$

$$-\gamma \left(\frac{z\left(\mathcal{I}_{n,m}^{\lambda,\alpha}f(z) - \mathcal{I}_{n,m}^{\lambda,\alpha}f(-z)\right)'}{\mathcal{I}_{n,m}^{\lambda,\alpha}f(z) - \mathcal{I}_{n,m}^{\lambda,\alpha}f(-z)}\right)\left(\frac{2z}{\mathcal{I}_{n,m}^{\lambda,\alpha}f(z) - \mathcal{I}_{n,m}^{\lambda,\alpha}f(-z)}\right)^{\mu} \text{ is univalent in } \mathbb{U}.$$

*If*

$$q(z) + \frac{\gamma}{\mu}zq'(z) \prec (1+\gamma)\left(\frac{2z}{\mathcal{I}_{n,m}^{\lambda,\alpha}f(z) - \mathcal{I}_{n,m}^{\lambda,\alpha}f(-z)}\right)^{\mu}$$

$$-\gamma \left(\frac{z\left(\mathcal{I}_{n,m}^{\lambda,\alpha}f(z) - \mathcal{I}_{n,m}^{\lambda,\alpha}f(-z)\right)'}{\mathcal{I}_{n,m}^{\lambda,\alpha}f(z) - \mathcal{I}_{n,m}^{\lambda,\alpha}f(-z)}\right)\left(\frac{2z}{\mathcal{I}_{n,m}^{\lambda,\alpha}f(z) - \mathcal{I}_{n,m}^{\lambda,\alpha}f(-z)}\right)^{\mu},$$

*then*

$$q(z) \prec \left(\frac{2z}{\mathcal{I}_{n,m}^{\lambda,\alpha}f(z) - \mathcal{I}_{n,m}^{\lambda,\alpha}f(-z)}\right)^{\mu},$$

*and $q$ is the best subordinant of Equation (25).*

Combining Theorems 3 and 4, we obtain the following sandwich-type theorem:

**Theorem 5.** *Let $q_1$ and $q_2$ be two convex functions in $\mathbb{U}$, with $q_1(0) = q_2(0) = 1$, and let $\gamma \in \mathbb{C}^*$, with $\mathrm{Re}\,\gamma \geq 0$. If $f \in \mathcal{A}(m)$ such that the assumptions in Equations (23) and (24) hold, then*

$$q_1(z) + \frac{\gamma}{\mu}zq_1'(z) \prec \Theta(z) := (1+\gamma)\left(\frac{2z}{\mathcal{N}_{n,m,q}^{\lambda,\alpha}f(z) - \mathcal{N}_{n,m,q}^{\lambda,\alpha}f(-z)}\right)^{\mu}$$

$$-\gamma\left(\frac{z\left(\mathcal{N}_{n,m,q}^{\lambda,\alpha}f(z) - \mathcal{N}_{n,m,q}^{\lambda,\alpha}f(-z)\right)'}{\mathcal{N}_{n,m,q}^{\lambda,\alpha}f(z) - \mathcal{N}_{n,m,q}^{\lambda,\alpha}f(-z)}\right)\left(\frac{2z}{\mathcal{N}_{n,m,q}^{\lambda,\alpha}f(z) - \mathcal{N}_{n,m,q}^{\lambda,\alpha}f(-z)}\right)^{\mu} \tag{27}$$

$$\prec q_2(z) + \frac{\gamma}{\mu}zq_2'(z),$$

*implies that*

$$q_1(z) \prec \Phi(z) := \left(\frac{2z}{\mathcal{N}_{n,m,q}^{\lambda,\alpha}f(z) - \mathcal{N}_{n,m,q}^{\lambda,\alpha}f(-z)}\right)^{\mu} \prec q_2(z),$$

*and $q_1$ and $q_2$ are, respectively, the best subordinant and the best dominant of Equation (27).*

Combining Corollaries 4 and 6, we obtain the following sandwich-type theorem:

**Corollary 7.** *Let $q_1$ and $q_2$ be two convex functions in $\mathbb{U}$, with $q_1(0) = q_2(0) = 1$, and let $\gamma \in \mathbb{C}^*$, with $\mathrm{Re}\,\gamma \geq 0$. If $f \in \mathcal{A}(m)$ such that the assumptions in Equations (23) and (24) hold for the operator $\mathcal{N}_{n,m,q}^{\lambda,\alpha}$ replaced by $\mathcal{I}_{n,m,q}^{\lambda,\alpha}$, then*

$$q_1(z) + \frac{\gamma}{\mu}zq_1'(z) \prec \widehat{\Theta}(z) := (1+\gamma)\left(\frac{2z}{\mathcal{I}_{n,m}^{\lambda,\alpha}f(z) - \mathcal{I}_{n,m}^{\lambda,\alpha}f(-z)}\right)^{\mu}$$

$$-\gamma\left(\frac{z\left(\mathcal{I}_{n,m}^{\lambda,\alpha}f(z) - \mathcal{I}_{n,m}^{\lambda,\alpha}f(-z)\right)'}{\mathcal{I}_{n,m}^{\lambda,\alpha}f(z) - \mathcal{I}_{n,m}^{\lambda,\alpha}f(-z)}\right)\left(\frac{2z}{\mathcal{I}_{n,m}^{\lambda,\alpha}f(z) - \mathcal{I}_{n,m}^{\lambda,\alpha}f(-z)}\right)^{\mu} \prec q_2(z) + \frac{\gamma}{\mu}zq_2'(z), \tag{28}$$

*implies that*

$$q_1(z) \prec \widehat{\Phi}(z) := \left(\frac{2z}{\mathcal{I}_{n,m}^{\lambda,\alpha}f(z) - \mathcal{I}_{n,m}^{\lambda,\alpha}f(-z)}\right)^{\mu} \prec q_2(z),$$

*and $q_1$ and $q_2$ are, respectively, the best subordinant and the best dominant of Equation (27).*

**Example 2.** *Taking $q_j = 1 + r_j z$, with $0 < r_1 < r_2$, $j = 1, 2$ in Theorem 5 and Corollary 7, we obtain the next examples, respectively:*

*Let $\gamma \in \mathbb{C}^*$, with $\mathrm{Re}\,\gamma \geq 0$.*

*1. If $f \in \mathcal{A}(m)$ such that the assumptions in Equations (23) and (24) hold, then*

$$r_1\left|1 + \frac{\gamma}{\mu}\right| < |\Theta(z) - 1| < r_2\left|1 + \frac{\gamma}{\mu}\right|, \ z \in \mathbb{U} \Rightarrow r_1 < |\Phi(z) - 1| < r_2, \ z \in \mathbb{U}, \quad (0 < r_1 < r_2)$$

*where $\Theta$ and $\Phi$ are given in Theorem 5, and the obtained bounds $r_1$ and $r_2$ are the best possible.*

*2. If $f \in \mathcal{A}(m)$ such that the assumptions in Equations (23) and (24) hold for the operator $\mathcal{N}_{n,m,q}^{\lambda,\alpha}$ replaced by $\mathcal{I}_{n,m,q}^{\lambda,\alpha}$, then*

$$r_1\left|1 + \frac{\gamma}{\mu}\right| < |\widehat{\Theta}(z) - 1| < r_2\left|1 + \frac{\gamma}{\mu}\right|, \ z \in \mathbb{U} \Rightarrow r_1 < |\widehat{\Phi}(z) - 1| < r_2, \ z \in \mathbb{U}, \quad (0 < r_1 < r_2)$$

*where $\widehat{\Theta}$ and $\widehat{\Phi}$ are given in Corollary 7, and the obtained bounds $r_1$ and $r_2$ are the best possible.*

**Example 3.** *Putting $q_j = e^{r_j z}$, with $0 < r_1 < r_2 \leq 1$, $j = 1, 2$ in Theorem 5 and Corollary 7, we obtain the next examples, respectively:*

*Let $\gamma \in \mathbb{C}^*$, with $\mathrm{Re}\,\gamma \geq 0$.*

　　*1. If $f \in \mathcal{A}(m)$ such that the assumptions in Equations (23) and (24) hold, then*

$$\left(1 + \frac{\gamma}{\mu}z\right) e^{r_1 z} \prec \Theta(z) \prec \left(1 + \frac{\gamma}{\mu}z\right) e^{r_2 z} \Rightarrow e^{r_1 z} \prec \Phi(z) \prec e^{r_2 z}, \quad (0 < r_1 < r_2 \le 1)$$

*where $\Theta$ and $\Phi$ are given in Theorem 5, and $e^{r_1 z}$ and $e^{r_2 z}$ are, respectively, the best subordinant and the best dominant.*

　　*2. If $f \in \mathcal{A}(m)$ such that the assumptions in Equations (23) and (24) hold for the operator $\mathcal{N}_{n,m,q}^{\lambda,\alpha}$ replaced by $\mathcal{I}_{n,m,q}^{\lambda,\alpha}$, then*

$$\left(1 + \frac{\gamma}{\mu}z\right) e^{r_1 z} \prec \widehat{\Theta}(z) \prec \left(1 + \frac{\gamma}{\mu}z\right) e^{r_2 z} \Rightarrow e^{r_1 z} \prec \widehat{\Phi}(z) \prec e^{r_2 z}, \quad (0 < r_1 < r_2 \le 1)$$

*where $\widehat{\Theta}$ and $\widehat{\Phi}$ are given in Corollary 7, and $e^{r_1 z}$ and $e^{r_2 z}$ are, respectively, the best subordinant and the best dominant.*

**Theorem 6.** *If $f \in \mathcal{M}_{n,m,q}^{\lambda,\alpha}(0, \mu, 1 - 2\rho, -1)$, with $0 \le \rho < 1$, then $f \in \mathcal{M}_{n,m,q}^{\lambda,\alpha}(\gamma, \mu, 1 - 2\rho, -1)$ for $|z| < R$, where*

$$R = \left(\sqrt{\frac{|\gamma|^2 m^2}{\mu^2} + 1} - \frac{|\gamma| m}{\mu}\right)^{\frac{1}{m}}. \tag{29}$$

**Proof.** For $f \in \mathcal{M}_{n,m,q}^{\lambda,\alpha}(0, \mu, 1 - 2\rho, -1)$, with $0 \le \rho < 1$, let the function $h$ be defined by

$$\left(\frac{2z}{\mathcal{N}_{n,m,q}^{\lambda,\alpha} f(z) - \mathcal{N}_{n,m,q}^{\lambda,\alpha} f(-z)}\right)^{\mu} = (1 - \rho)h(z) + \rho, \; z \in \mathbb{U}. \tag{30}$$

　　Hence, the function $h$ is analytic in $\mathbb{U}$, with $h(0) = 1$, and since $f \in \mathcal{M}_{n,m,q}^{\lambda,\alpha}(0, \mu, 1 - 2\rho, -1)$ is equivalent to,

$$\left(\frac{2z}{\mathcal{N}_{n,m,q}^{\lambda,\alpha} f(z) - \mathcal{N}_{n,m,q}^{\lambda,\alpha} f(-z)}\right)^{\mu} \prec \frac{1 + (1 - 2\rho)z}{1 - z},$$

it follows that $\operatorname{Re} h(z) > 0, z \in \mathbb{U}$.

　　As in the proof of Theorem 1, since $f \in \mathcal{M}_{n,m,q}^{\lambda,\alpha}(0, \mu, 1 - 2\rho, -1)$, with $0 \le \rho < 1$, we deduce that

$$\left(\frac{2z}{\mathcal{N}_{n,m,q}^{\lambda,\alpha} f(z) - \mathcal{N}_{n,m,q}^{\lambda,\alpha} f(-z)}\right)^{\mu} \in \mathcal{H}[1, m],$$

and from the relation in Equation (30), we get $h \in \mathcal{H}[1, m]$. Therefore, the following estimate holds

$$\left|zh'(z)\right| \le \frac{2mr^m \operatorname{Re} h(z)}{1 - r^{2m}}, \; |z| = r < 1,$$

that represents the result of Shah [19] (the inequality (6), p. 240, for $\alpha = 0$), which generalize Lemma 2 of [20].

A simple computation shows that

$$
\frac{1}{1-\rho}\left\{(1+\gamma)\left(\frac{2z}{\mathcal{N}_{n,m,q}^{\lambda,\alpha}f(z)-\mathcal{N}_{n,m,q}^{\lambda,\alpha}f(-z)}\right)^{\mu}\right.
$$

$$
\left.-\gamma\left(\frac{z\left(\mathcal{N}_{n,m,q}^{\lambda,\alpha}f(z)-\mathcal{N}_{n,m,q}^{\lambda,\alpha}f(-z)\right)'}{\mathcal{N}_{n,m,q}^{\lambda,\alpha}f(z)-\mathcal{N}_{n,m,q}^{\lambda,\alpha}f(-z)}\right)\left(\frac{2z}{\mathcal{N}_{n,m,q}^{\lambda,\alpha}f(z)-\mathcal{N}_{n,m,q}^{\lambda,\alpha}f(-z)}\right)^{\mu}-\rho\right\}
$$

$$
= h(z)+\frac{\gamma}{\mu}zh'(z),\ z\in\mathbb{U},
$$

hence, we obtain

$$
\mathrm{Re}\left\{\frac{1}{1-\rho}\left[(1+\gamma)\left(\frac{2z}{\mathcal{N}_{n,m,q}^{\lambda,\alpha}f(z)-\mathcal{N}_{n,m,q}^{\lambda,\alpha}f(-z)}\right)^{\mu}\right.\right.
$$

$$
\left.\left.-\gamma\left(\frac{z\left(\mathcal{N}_{n,m,q}^{\lambda,\alpha}f(z)-\mathcal{N}_{n,m,q}^{\lambda,\alpha}f(-z)\right)'}{\mathcal{N}_{n,m,q}^{\lambda,\alpha}f(z)-\mathcal{N}_{n,m,q}^{\lambda,\alpha}f(-z)}\right)\left(\frac{2z}{\mathcal{N}_{n,m,q}^{\lambda,\alpha}f(z)-\mathcal{N}_{n,m,q}^{\lambda,\alpha}f(-z)}\right)^{\mu}-\rho\right]\right\} \tag{31}
$$

$$
\geq\mathrm{Re}\,h(z)\left[1-\frac{2|\gamma|mr^{m}}{\mu\left(1-r^{2m}\right)}\right],\ |z|=r<1,
$$

and the right-hand side of Equation (31) is positive provided that $r < R$, where $R$ is given by Equation (29). $\square$

**Theorem 7.** *Let* $f\in\mathcal{M}_{n,m,q}^{\lambda,\alpha}(\gamma,\mu,A,B)$, *let* $\gamma\in\mathbb{C}^{*}$ *with* $\mathrm{Re}\,\gamma\geq 0$, *and* $-1\leq B<A\leq 1$.
   1. *Then,*

$$
\frac{\mu}{\gamma m}\int_{0}^{1}\frac{1-Au}{1-Bu}u^{\frac{\mu}{\gamma m}-1}du < \mathrm{Re}\left(\frac{2z}{\mathcal{N}_{n,m,q}^{\lambda,\alpha}f(z)-\mathcal{N}_{n,m,q}^{\lambda,\alpha}f(-z)}\right)^{\mu}
$$

$$
< \frac{\mu}{\gamma m}\int_{0}^{1}\frac{1+Au}{1+Bu}u^{\frac{\mu}{\gamma m}-1}du,\ z\in\mathbb{U}. \tag{32}
$$

   2. *For* $|z|=r<1$, *we have*

$$
2r\left(\frac{\mu}{\gamma m}\int_{0}^{1}\frac{1+Aur}{1+Bur}u^{\frac{\mu}{\gamma m}-1}du\right)^{-\frac{1}{\mu}} < \left|\mathcal{N}_{n,m,q}^{\lambda,\alpha}f(z)-\mathcal{N}_{n,m,q}^{\lambda,\alpha}f(-z)\right|
$$

$$
< 2r\left(\frac{\mu}{\gamma m}\int_{0}^{1}\frac{1-Aur}{1-Bur}u^{\frac{\mu}{\gamma m}-1}du\right)^{-\frac{1}{\mu}}. \tag{33}
$$

*All these inequalities are the best possible.*

**Proof.** From the assumptions, using Theorem 1, we obtain that

$$
\left(\frac{2z}{\mathcal{N}_{n,m,q}^{\lambda,\alpha}f(z)-\mathcal{N}_{n,m,q}^{\lambda,\alpha}f(-z)}\right)^{\mu} \prec \Psi(z) := \frac{\mu}{\gamma m}\int_{0}^{1}\frac{1+Azu}{1+Bzu}u^{\frac{\mu}{\gamma m}-1}du, \tag{34}
$$

and the convex function $\Psi\in\mathcal{H}[1,m]$ is the best dominant. Therefore,

$$
\mathrm{Re}\left(\frac{2z}{\mathcal{N}_{n,m,q}^{\lambda,\alpha}f(z)-\mathcal{N}_{n,m,q}^{\lambda,\alpha}f(-z)}\right)^{\mu} < \sup_{z\in\mathbb{U}}\mathrm{Re}\left(\frac{\mu}{\gamma m}\int_{0}^{1}\frac{1+Azu}{1+Bzu}u^{\frac{\mu}{\gamma m}-1}du\right)
$$

$$
= \frac{\mu}{\gamma m}\int_{0}^{1}\sup_{z\in\mathbb{U}}\mathrm{Re}\left(\frac{1+Azu}{1+Bzu}\right)u^{\frac{\mu}{\gamma m}-1}du = \frac{\mu}{\gamma m}\int_{0}^{1}\frac{1+Au}{1+Bu}u^{\frac{\mu}{\gamma m}-1}du,\ z\in\mathbb{U},
$$

and

$$\mathrm{Re}\left(\frac{2z}{\mathcal{N}_{n,m,q}^{\lambda,\alpha}f(z)-\mathcal{N}_{n,m,q}^{\lambda,\alpha}f(-z)}\right)^{\mu} > \inf_{z\in\mathbb{U}}\mathrm{Re}\left(\frac{\mu}{\gamma m}\int_0^1 \frac{1-Azu}{1-Bzu}u^{\frac{\mu}{\gamma m}-1}du\right)$$

$$=\frac{\mu}{\gamma m}\int_0^1 \inf_{z\in\mathbb{U}}\mathrm{Re}\left(\frac{1-Azu}{1-Bzu}\right)u^{\frac{\mu}{\gamma m}-1}du = \frac{\mu}{\gamma m}\int_0^1 \frac{1-Au}{1-Bu}u^{\frac{\mu}{\gamma m}-1}du,\ z\in\mathbb{U}.$$

In addition, since

$$\left|\frac{2z}{\mathcal{N}_{n,m,q}^{\lambda,\alpha}f(z)-\mathcal{N}_{n,m,q}^{\lambda,\alpha}f(-z)}\right|^{\mu} < \sup_{z\in\mathbb{U}}\left|\frac{\mu}{\gamma m}\int_0^1 \frac{1+Azu}{1+Bzu}u^{\frac{\mu}{\gamma m}-1}du\right|$$

$$=\frac{\mu}{\gamma m}\int_0^1 \sup_{z\in\mathbb{U}}\left|\frac{1+Azu}{1+Bzu}\right|u^{\frac{\mu}{\gamma m}-1}du = \frac{\mu}{\gamma m}\int_0^1 \frac{1+Aur}{1+Bur}u^{\frac{\mu}{\gamma m}-1}du,\ |z|=r<1,$$

we get

$$\left|\mathcal{N}_{n,m,q}^{\lambda,\alpha}f(z)-\mathcal{N}_{n,m,q}^{\lambda,\alpha}f(-z)\right| > 2r\left(\frac{\mu}{\gamma m}\int_0^1 \frac{1+Aur}{1+Bur}u^{\frac{\mu}{\gamma m}-1}du\right)^{-\frac{1}{\mu}},$$

while

$$\left|\frac{2z}{\mathcal{N}_{n,m,q}^{\lambda,\alpha}f(z)-\mathcal{N}_{n,m,q}^{\lambda,\alpha}f(-z)}\right|^{\mu} > \inf_{z\in\mathbb{U}}\left|\frac{\mu}{\gamma m}\int_0^1 \frac{1-Azu}{1-Bzu}u^{\frac{\mu}{\gamma m}-1}du\right|$$

$$=\frac{\mu}{\gamma m}\int_0^1 \inf_{z\in\mathbb{U}}\left|\frac{1-Azu}{1-Bzu}\right|u^{\frac{\mu}{\gamma m}-1}du = \frac{\mu}{\gamma m}\int_0^1 \frac{1-Aur}{1-Bur}u^{\frac{\mu}{\gamma m}-1}du,\ |z|=r<1,$$

implies

$$\left|\mathcal{N}_{n,m,q}^{\lambda,\alpha}f(z)-\mathcal{N}_{n,m,q}^{\lambda,\alpha}f(-z)\right| < 2r\left(\frac{\mu}{\gamma m}\int_0^1 \frac{1-Aur}{1-Bur}u^{\frac{\mu}{\gamma m}-1}du\right)^{-\frac{1}{\mu}}.$$

The inequalities of Equations (32) and (33) are the best possible because the subordination in Equation (34) is sharp. □

Taking $q\to 1^-$ in Theorem 7, we obtain the following corollary:

**Corollary 8.** *Let $f\in\mathcal{W}_{n,m}^{\lambda,\alpha}(\gamma,\mu,A,B)$, let $\gamma\in\mathbb{C}^*$ with $\mathrm{Re}\,\gamma\geq 0$, and $-1\leq B<A\leq 1$.*
*1. Then,*

$$\frac{\mu}{\gamma m}\int_0^1 \frac{1-Au}{1-Bu}u^{\frac{\mu}{\gamma m}-1}du < \mathrm{Re}\left(\frac{2z}{\mathcal{I}_{n,m}^{\lambda,\alpha}f(z)-\mathcal{I}_{n,m}^{\lambda,\alpha}f(-z)}\right)^{\mu}$$

$$< \frac{\mu}{\gamma m}\int_0^1 \frac{1+Au}{1+Bu}u^{\frac{\mu}{\gamma m}-1}du,\ z\in\mathbb{U}.$$

*2. For $|z| = r < 1$, we have*

$$2r \left( \frac{\mu}{\gamma m} \int_0^1 \frac{1 + Aur}{1 + Bur} u^{\frac{\mu}{\gamma m} - 1} du \right)^{-\frac{1}{\mu}} < \left| \mathcal{I}_{n,m}^{\lambda,\alpha} f(z) - \mathcal{I}_{n,m}^{\lambda,\alpha} f(-z) \right|$$

$$< 2r \left( \frac{\mu}{\gamma m} \int_0^1 \frac{1 - Aur}{1 - Bur} u^{\frac{\mu}{\gamma m} - 1} du \right)^{-\frac{1}{\mu}}.$$

*All these inequalities are the best possible.*

Taking $q \to 1^-$, $\alpha = 0$ and $\lambda = 1$ in Theorem 7, we obtain the following corollary:

**Corollary 9.** *Let $f \in \mathcal{N}^{\gamma,\mu}(m, A, B)$, let $\gamma \in \mathbb{C}^*$ with $\operatorname{Re} \gamma \geq 0$, and $-1 \leq B < A \leq 1$.*
*1. Then,*

$$\frac{\mu}{\gamma m} \int_0^1 \frac{1 - Au}{1 - Bu} u^{\frac{\mu}{\gamma m} - 1} du < \operatorname{Re} \left( \frac{2z}{f(z) - f(-z)} \right)^{\mu} < \frac{\mu}{\gamma m} \int_0^1 \frac{1 + Au}{1 + Bu} u^{\frac{\mu}{\gamma m} - 1} du, \ z \in \mathbb{U}.$$

*2. For $|z| = r < 1$, we have*

$$2r \left( \frac{\mu}{\gamma m} \int_0^1 \frac{1 + Aur}{1 + Bur} u^{\frac{\mu}{\gamma m} - 1} du \right)^{-\frac{1}{\mu}} < |f(z) - f(-z)|$$

$$< 2r \left( \frac{\mu}{\gamma m} \int_0^1 \frac{1 - Aur}{1 - Bur} u^{\frac{\mu}{\gamma m} - 1} du \right)^{-\frac{1}{\mu}}.$$

*All these inequalities are the best possible.*

**Example 4.** *Putting $\mu = \gamma = m = 1$, $A = 1 - 2\beta$ $(0 \leq \beta < 1)$, and $B = -1$ in Corollary 9, we get the next special case.*
*If $f \in \mathcal{N}^{1,1}(1, 1 - 2\beta, -1)$ with $0 \leq \beta < 1$, then:*
*1. The next inequality holds:*

$$\operatorname{Re} \frac{2z}{f(z) - f(-z)} > 2\beta - 1 + 2(1 - \beta) \ln 2, \ z \in \mathbb{U}.$$

*2. For $|z| = r := 0.9$, we have*

$$\frac{1.8}{1 + 3.116855762\beta} < |f(z) - f(-z)| < \frac{1.8}{1 - 0.5736580307\beta}.$$

**Remark 2.** *Part (ii) of Corollary 9 corrects the Corollary (3.10) studied by Muhammad and Marwan [16].*

Concluding, all the above results give us information about subordination and superordination properties, inclusion results, radius problem, and sharp estimations for the classes $\mathcal{M}_{n,m,q}^{\lambda,\alpha}(\gamma, \mu, A, B)$, together general sharp subordination and superordination for the operator $\mathcal{N}_{n,m,q}^{\lambda,\alpha}$. For special choices of the parameters $\gamma \in \mathbb{C}$, $0 < \mu < 1$, $-1 \leq B < A \leq 1$, $m \in \mathbb{N}$, $\alpha > 0$, $n \geq 0$, $0 < q < 1$, and $\lambda > -1$, we may obtain several simple applications connected with the above-mentioned classes and operator.

**Author Contributions:** The authors contributed equally to this work.

**Funding:** This research received no external funding.

**Acknowledgments:** The authors are grateful to the reviewers of this article, who gave valuable remarks, comments, and advice, in order to revise and improve the results of the paper.

**Conflicts of Interest:** The authors declare no conflict of interest.

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
