# Peer review of "Differential Sandwich-Type Results for Symmetric Functions Connected with a Q-Analog Integral Operator"

_mathematics, doi:10.3390/math7121185_

Round 1

Reviewer 1 Report

The authors study differential subordination of complex functions and present
several sandwich-type theorems.

The introduction introduces well the main concepts of the paper, but it has
a few grammatical mistakes. Maybe the authors want to use the more common
notation (q;q)_n for q-Pochhammer symbols.

The main part is quite technical, but looks solid. However, I'm not able to
check the correctness of the results, neither can I tell whether this research
is new or not. Concerning language and typsetting the manuscript has been
carefully prepared.

In line 51, the authors should replace "reminder" by "remainder".

Therefore, this report does not come with a clear recommendation, whether to
accept or reject the paper.

Author Response

Thank you for your comments, that help us to improve the quality of our work. We will take into the account your recommendations, and we will upload soon the revised form of the paper, according to the recommendations of all the three reviewers.

The authors.

Reviewer 2 Report

The present paper deals with some applications of the theory of differential subordination, differential superordination, and sandwich-type results for some sub-classes of symmetric functions connected with a q-analogue integral operator.

On one hand, the q-infinitesimal calculus can have many potential applications in mathematics and (perhaps) in physics. On the other hand, there is no mention concerning such possible applications. In conclusion, the paper is an honest one concerning some exotic argument, whose (dubious) soundness and interest is not explained at all.

However, the paper appears well written and, after a quick look, the results appear correct. Taking also into account that the journal is not against such kind of works, the referee conforms himself with such an approach.

Author Response

(The authors gave the same response as above.)

Reviewer 3 Report

The manuscript is aimed at analyzing di erential subordination and
differential superordination for special classes of functions and operators.
The manuscript is well-organized, the methodology is clear, the background theory is reviewed and well-exposed, the results are ordered
and clear, the Abstract and Concluding Remarks
Section contain the main element of the manuscript, the bibliography is almost rather complete.

Overall editorial suggestion: Major revisions.
Major revisions:
To improve the manuscript, further applications could be presented as examples.

Author Response

(The authors gave the same response as above.)

Round 2

Reviewer 3 Report

The Authors have improved the manuscript by adding several interesting examples, thus augmenting the effectiveness and the profitability of the manuscript.

Overall Editorial Suggestion: publication.